UPDATE ARTICLE

# Behavioral adjustment of *C. elegans* to mechanosensory loss requires intact mechanosensory neurons

**Michal Staum, Ayelet-Chen Abraham, Reema Arbid, Varun Sanjay Birari, Matanel Dominitz, Ithai Rabinowitch** [ORCID] *

Department of Medical Neurobiology, Institute for Medical Research Israel-Canada, Faculty of Medicine, Hebrew University of Jerusalem, Jerusalem, Israel

* ithai.rabinowitch@mail.huji.ac.il

The Editors encourage authors to publish research updates to this article type. Please follow the link in the citation below to view any related articles.

## Abstract

Sensory neurons specialize in detecting and signaling the presence of diverse environmental stimuli. Neuronal injury or disease may undermine such signaling, diminishing the availability of crucial information. Can animals distinguish between a stimulus not being present and the inability to sense that stimulus in the first place? To address this question, we studied *Caenorhabditis elegans* nematode worms that lack gentle body touch sensation due to genetic mechanoreceptor dysfunction. We previously showed that worms can compensate for the loss of touch by enhancing their sense of smell, via an FLP-20 neuropeptide pathway. Here, we find that touch-deficient worms exhibit, in addition to sensory compensation, also cautious-like behavior, as if preemptively avoiding potential undetectable hazards. Intriguingly, these behavioral adjustments are abolished when the touch neurons are removed, suggesting that touch neurons are required for signaling the unavailability of touch information, in addition to their conventional role of signaling touch stimulation. Furthermore, we found that the ASE taste neurons, which similarly to the touch neurons, express the FLP-20 neuropeptide, exhibit altered FLP-20 expression levels in a touch-dependent manner, thus cooperating with the touch circuit. These results imply a novel form of neuronal signaling that enables *C. elegans* to distinguish between lack of touch stimulation and loss of touch sensation, producing adaptive behavioral adjustments that could overcome the inability to detect potential threats.

## Introduction

Animals use sensory information to locate useful resources in the environment and to avoid dangers. This is enabled by distinct sensory neurons specialized in detecting particular environmental signals. For example, olfactory sensory neurons detecting favorable food odors may direct attention and locomotion towards these odor sources, and mechanosensory neurons responding to a sudden touch may drive a defensive behavior, such as a lash or a withdrawal.

**Data Availability Statement:** All relevant data are within the paper and its Supporting Information files. Custom code used in the study is available at Github at the following locations: https://github.com/RabinowitchLab/RecordGUI.git (10.5281/zenodo.12193290) https://github.com/RabinowitchLab/TREX.git (10.5281/zenodo.12193275) https://github.com/RabinowitchLab/NeuroShine.git (10.5281/zenodo.12193342).

**Funding:** IR received funding for this study from the Israel Science Foundation (https://www.isf.org.il) Grant No. 1465/20 and from Horizon Europe, PathFinder European Innovation Council (https://eic.ec.europa.eu/eic-funding-opportunities/eic-pathfinder_en) Grant No. 101098722. Neither funder played any role in the study design, data collection and analysis, decision to publish, or preparation of the manuscript.

**Competing interests:** The authors have declared that no competing interests exist.

**Abbreviations:** CI, chemotaxis index; NGM, nematode growth medium; NP, neuropeptide; ROI, region of interest; TRN, touch receptor neuron.

Certain disease conditions or injuries may cause sensory loss, such as blindness or deafness. Remarkably, in many such cases the nervous system is able to compensate for the sensory deficit through cross-modal plasticity [1,2], whereby some of the missing information is provided by remaining senses whose performance is modulated. This effect has been broadly demonstrated in humans, such as in blind people exhibiting increased auditory [3], touch [4], and olfactory [5] acuity, and in many other animals, including even the 1-mm long nematode worm *Caenorhabditis elegans* [1,6,7] whose nervous system consists of only several hundred neurons. Cross-modal plasticity may thus help circumvent the loss of sensory information, making it possible to successfully navigate the environment despite the absence of certain sensory inputs.

Nevertheless, certain environmental signals may be exclusive to one specific modality, and thus cannot be substituted by the modulation of remaining sensory modalities. This is especially consequential for the detection of potential dangers, for example, a red traffic light (vision) or the smell of gas (olfaction). How do animals adjust to an inability to sense potential hazards, and what are the roles of the sensory neurons that no longer convey sensory information in such conditions? To address these questions, we studied various defensive-like behaviors of *C. elegans* worms deprived of body touch sensation, and specifically examined the adjustment to loss of touch in the absence of the touch receptor neurons (TRNs), as opposed to mere touch receptor dysfunction.

## Results

### Behavioral adjustment following loss of touch requires intact touch neurons

We have previously shown that loss of touch sensation leads to behavioral adjustments in *C. elegans*, including enhanced spontaneous reversing upon food removal [6]. These effects occur in mechanoreceptor mutants, such as *mec-4(u253)* (henceforth, *mec-4*), which lack a functional DEG/ENaC (degenerin/epithelial Na$^+$ channel) subunit, necessary for sensing gentle touch to the body [8,9], but still possess intact TRNs (Fig 1A and 1B). *mec-4* is expressed specifically in the TRNs [10–12]. We expected that removal of the TRNs would result in similar if not greater behavioral adjustment compared to merely eliminating mechanoreceptor function. To test this, we used the *mec-4(e1611)* allele (henceforth, *mec-4d*), which induces TRN death due to channel hyperactivity [13]. Surprisingly, unlike the enhanced reversing rate off food of *mec-4* mutants, *mec-4d* worms exhibited normal reversing (Fig 1B). A similar effect was observed in worms lacking touch neurons due to ectopic expression of the mammalian caspase gene, ICE [14], driven by the TRN-specific *Pmec-17* promoter (S1A Fig). Concomitantly, *Pmec-17*::*ICE* expressed in the *mec-4* mutant background fully suppressed the enhanced reversing of *mec-4* (S1B Fig), and when expressed in the *mec-4d* background showed no difference in reversing from *Pmec-17*::*ICE* or *mec-4d* alone (S1C Fig), confirming the effective elimination of the touch neurons in these worms.

Previously, we have also shown that enhanced odor-locomotion coupling of *mec-4* mutants leads to improved chemotaxis to food odors sensed by the AWC olfactory neurons [6] (Fig 1C). Worms lacking TRNs showed no such behavioral modification (Figs 1C and S1D). To reveal whether this compensatory behavior extends to additional modalities, we compared salt chemotaxis, mediated by the gustatory neurons, ASE, between N2, *mec-4* and *mec-4d* worms. In this case too, *mec-4* mutants showed enhanced gustatory acuity, whereas *mec-4d* showed no such sensory tuning (Figs 1D and S1E). These results suggest that the touch receptor neurons are necessary for compensatory behavioral adjustment following loss of touch sensation.

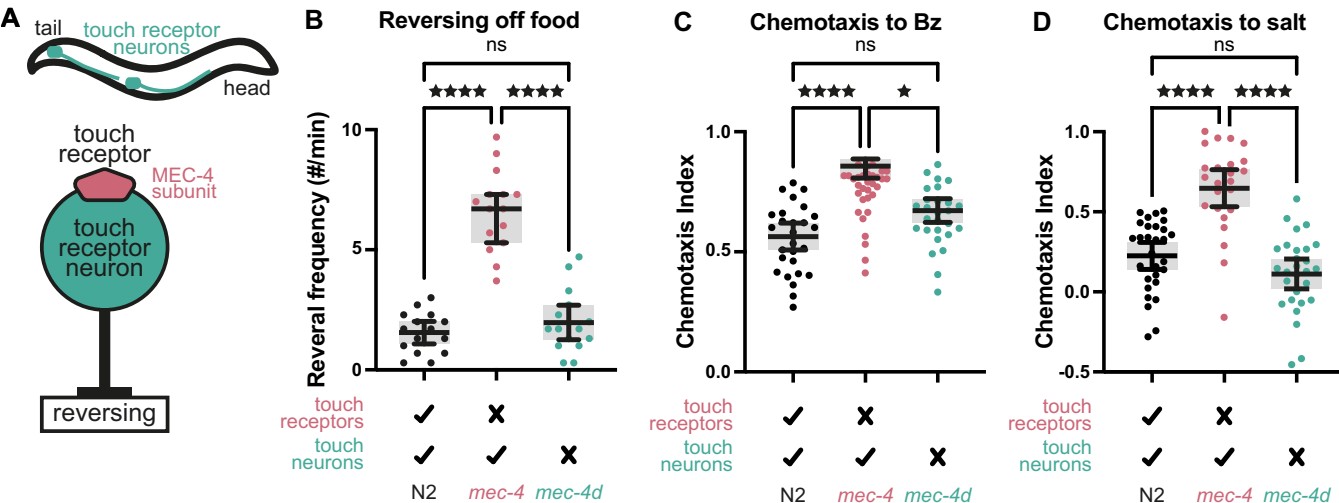

**Fig 1. Behavioral adjustment following loss of touch vs. loss of touch neurons.** (A) Schematic illustration of the *C. elegans* TRNs and their modulatory impact on ongoing locomotion. (B) Spontaneous reversing rate (# of reversals per minute) in the absence of food, for normal worms (N2), touch receptor mutants (*mec-4*), and worms lacking TRNs (*mec-4d*). One-way ANOVA ($p < 0.0001$) followed by Šídák's multiple comparisons test. ★★★★$p < 0.0001$. $n = 15$ for each condition. (C) Chemotaxis to 1:1,000 benzaldehyde (Bz) sensed by the AWC neurons. Kruskal–Wallis test ($p < 0.0001$) followed by Dunn's multiple comparisons test. ★$p < 0.05$, ★★★★$p < 0.0001$. $n = 27, 32, 27$. (D) Chemotaxis to 100 mM NaCl sensed by the ASE neurons. Kruskal–Wallis test ($p < 0.0001$) followed by Dunn's multiple comparisons test. ★$p < 0.05$, ★★★★$p < 0.0001$. $n = 29, 24, 28$. Bars indicate mean with 95% confidence intervals. The numerical data presented in this figure can be found in Supplementary file S1 Data.

### Loss of touch sensation elicits TRN-dependent restrictive-like behaviors

In addition to cross-modal adjustment, we wished to study the impact of loss of touch on defensive-like behaviors that may rely on touch information. First, we examined the trajectories of worms over a 2-min time window after removal from food (Fig 2A). We found that consistent with an enhanced reversing rate, *mec-4* trajectories were constrained relative to normal worms, as measured by a significantly reduced maximal radial distance from the point of origin (Fig 2A). A similar, although smaller behavioral modification was observed also in a separate touch-receptor mutant, *mec-10(e1515)* (S2A Fig). In contrast, *mec-4d* trajectories showed no such adjustment (Fig 2A), and neither did *Pmec-17::ICE* trajectories (S2B Fig). This finding suggests that in addition to cross-sensory compensation, such as enhanced olfactory and gustatory acuity (Fig 1C and 1D), loss of touch may lead also to restrictive locomotion, which could reduce, for example, the probability of encountering potential threats that are undetectable in the absence of touch sensation.

To further explore this possibility, we considered 2 additional presumably defensive behaviors. First, we studied worm escape responses to sudden touch to the nose, a behavior that is independent of the TRNs, and relies on other neurons [15] and mechanoreceptor genes [16,17]. As we have previously reported [6], *mec-4* worms exhibit a decreased nose touch response (Fig 2B). This behavioral change could be interpreted as a precautionary measure for avoiding mechanical obstacles and threats, which cannot be sensed by escaping *mec-4* worms. When testing *mec-4d* nose touch responses, we observed, in contrast, normal escape behavior (Fig 2B), consistent with a role for the TRNs in restricting behavioral responses when touch information is unavailable. This result also further confirms the dissociation between the nose touch response and the body touch circuit, as *mec-4d* mutants lacking body touch sensation exhibit normal escape from nose touch.

We also examined head oscillations, which are an intrinsic component of worm locomotion [18]. It has previously been shown that head oscillations are suppressed specifically during

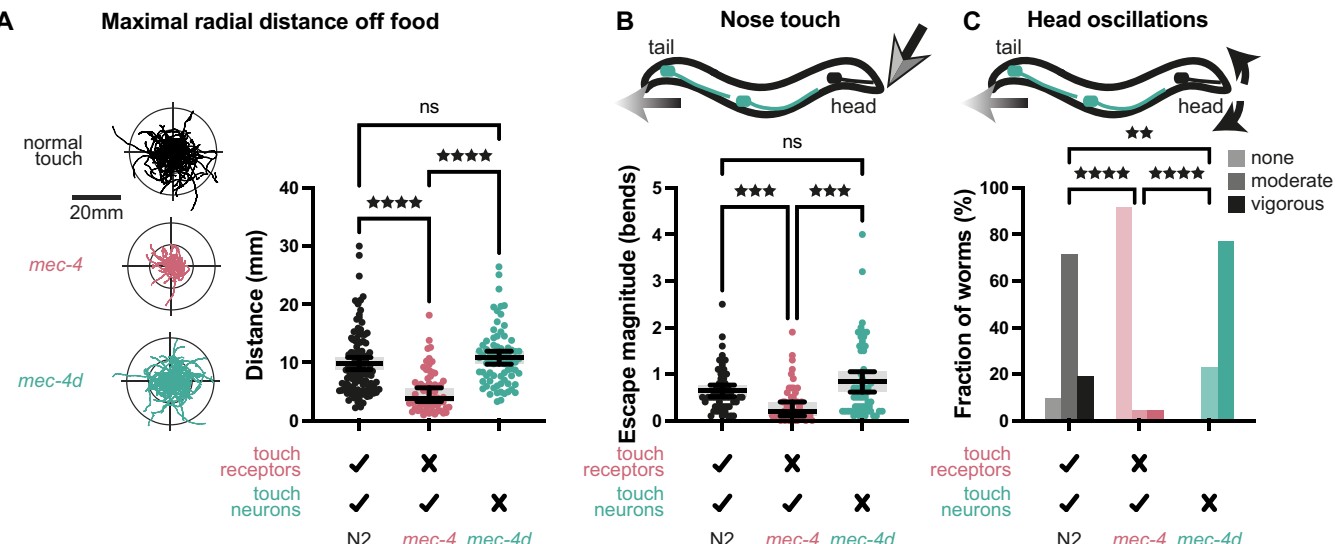

**Fig 2. Restrictive-like behaviors following loss of touch receptors vs. loss of touch neurons.** (A) Left, worm trajectories in the absence of food over a period of 2 min. The starting points of all trajectories are aligned to the same origin. Right, maximal radial distance for each trajectory from the starting point. See also S1 Fig. Kruskal–Wallis test ($p < 0.0001$) followed by Dunn's multiple comparisons test. ★★★★$p < 0.0001$. $n = 110, 70, 76$. (B) Escape response to gentle nose touch measured as number of body bends during the escape, for normal worms (N2), touch receptor mutants (*mec-4*), and worms lacking TRNs (*mec-4d*). Kruskal–Wallis test. ($p < 0.0001$) followed by Dunn's multiple comparisons test. ★★★$p < 0.001$. $n = 55, 53, 56$. (C) The extent of head oscillations during spontaneous reversing in the absence of food. The fraction of worms displaying vigorous, moderate or no oscillations are presented. Kolmogorov–Smirnov tests. ★★$p < 0.01$, ★★★★$p < 0.0001$. $n = 21, 24, 22$. Bars in A and B indicate mean with 95% confidence intervals. The numerical data presented in this figure can be found in Supplementary file S1 Data.

body touch-evoked reversing, as a possible strategy for avoiding predacious fungi [19]. We noticed that in contrast to normal worms, *mec-4* worms suppress head oscillations during spontaneous reversing, even though these reversals are not touch elicited (Fig 2C). Such *mec-4* suppression of head oscillations during reversing could preempt potential dangers that are normally associated with touch-evoked reversing. *mec-4d* worms, in contrast, showed no such suppression. To the contrary, they even exhibited exaggerated head oscillations (Fig 2C).

Together, these findings suggest that loss of touch sensation leads to both compensatory (Fig 1) and restrictive-like behaviors (Fig 2) that may reflect an adjustment to the unavailability of touch information. Moreover, the consistent lack of behavioral adjustment in *mec-4d* worms (Figs 1 and 2) suggests that the TRNs that normally convey touch information may signal the loss of touch when mechanosensation is compromised.

## Loss of touch is reflected in RIM interneuron activity

To appreciate how the differences between loss of touch sensation and loss of the touch neurons may be reflected in neuronal activity, we examined the motor neuron/interneuron RIM. This neuron plays important and opposing roles in regulating locomotion [20], depending on its activity patterns (evoked [18] versus ongoing [21]) and on the mode of synaptic interaction with its partners (chemical versus electrical [22,23]). We imaged RIM calcium activity over a period of 40 min in worms trapped in a locomotion microfluidic chip [24]. RIM recordings showed intermittent large increases in calcium concentration (Fig 3A). For each such event we calculated the percent change in fluorescence between the initiation of the event and its peak. Consistent with the behavioral results, *mec-4* RIM calcium activity showed significantly lower amplitudes than control (Fig 3B), while no such effect was evident for *mec-4d* mutants. Thus, loss of touch sensation alters RIM neuronal activity, but only as long as the TRNs are present.

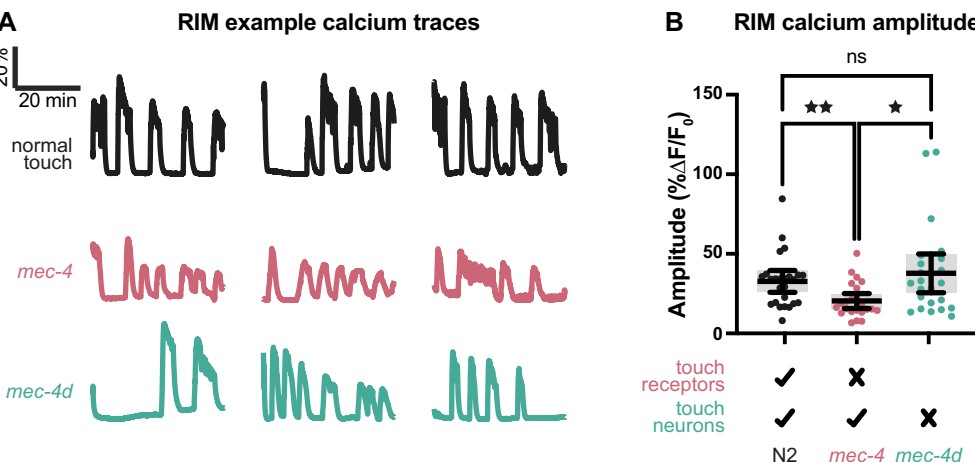

**Fig 3. RIM neuronal activity.** (A) Example traces of RIM activity over 40-min recording periods. (B) RIM calcium amplitudes. Kruskal–Wallis test ($p = 0.0027$) followed by Dunn's multiple comparisons test. ★$p < 0.01$, ★★$p < 0.01$. $n = 24, 23, 23$. Bars indicate mean with 95% confidence intervals. The numerical data presented in this figure can be found in Supplementary file S1 Data.

## FLP-20 neuropeptide levels differ between loss of touch and loss of touch neurons

Our previous studies [6] revealed that FLP-20 neuropeptide signaling from the TRNs plays an important role in modulating locomotion and chemosensation in the absence of touch sensation. In particular, loss of touch, such as occurs in *mec-4* mutants, leads to reduced FLP-20 secretion from the TRNs, increasing reversing off food [6]. We asked whether FLP-20 is important also for regulating dispersal behavior (Fig 4A). To test this, we examined 2 independent *flp-20* mutant alleles, *flp-20 (syb6941)* and *flp-20(ok2964)*, and found that similarly to *mec-4* worms, they too exhibit restricted dispersal (Figs 4B and S3A). Thus, reduced FLP-20 secretion in *mec-4* mutants could underlie the restricted dispersal of these touch-insensitive worms (Fig 4A, middle). On the face of it, *mec-4d* worms, completely lacking FLP-20 TRN signaling, should exhibit even more restricted dispersal. However, as we have found, dispersal remains unmodified in *mec-4d* mutants (Fig 2A).

This puzzling result could be resolved if unlike loss of touch per se, loss of the TRNs led to enhanced FLP-20 secretion from other neurons (Fig 4A, right). To test this possibility, we compared overall FLP-20 transcription levels between *mec-4* and *mec-4d* mutants relative to normal worms, using reverse transcription quantitative real-time PCR (RT-qPCR). Indeed, our results indicated significantly higher FLP-20 transcription in *mec-4d* compared to *mec-4* worms (Figs 4C and S3B), supporting our hypothesis that FLP-20 levels are higher in *mec-4d* compared to *mec-4* mutants.

We wondered whether FLP-20 signaling is involved in the modulation of RIM activity seen in worms insensitive to touch (Fig 3B). We found no significant difference in the average magnitude of RIM calcium events between *flp-20* mutants and control (Fig 4D), suggesting that the FLP-20 pathway involves other neurons, or that it influences aspects of RIM activity, such as downstream signaling, rather than calcium activity or that it acts redundantly with other signaling pathways.

## Elevated FLP-20 secretion from the ASE salt-sensing neurons eliminates behavioral adjustment upon TRN loss

What is the source of elevated FLP-20 secretion in *mec-4d* mutants? In addition to the TRNs, FLP-20 is expressed in several other neurons including the ASE gustatory salt-sensing neurons

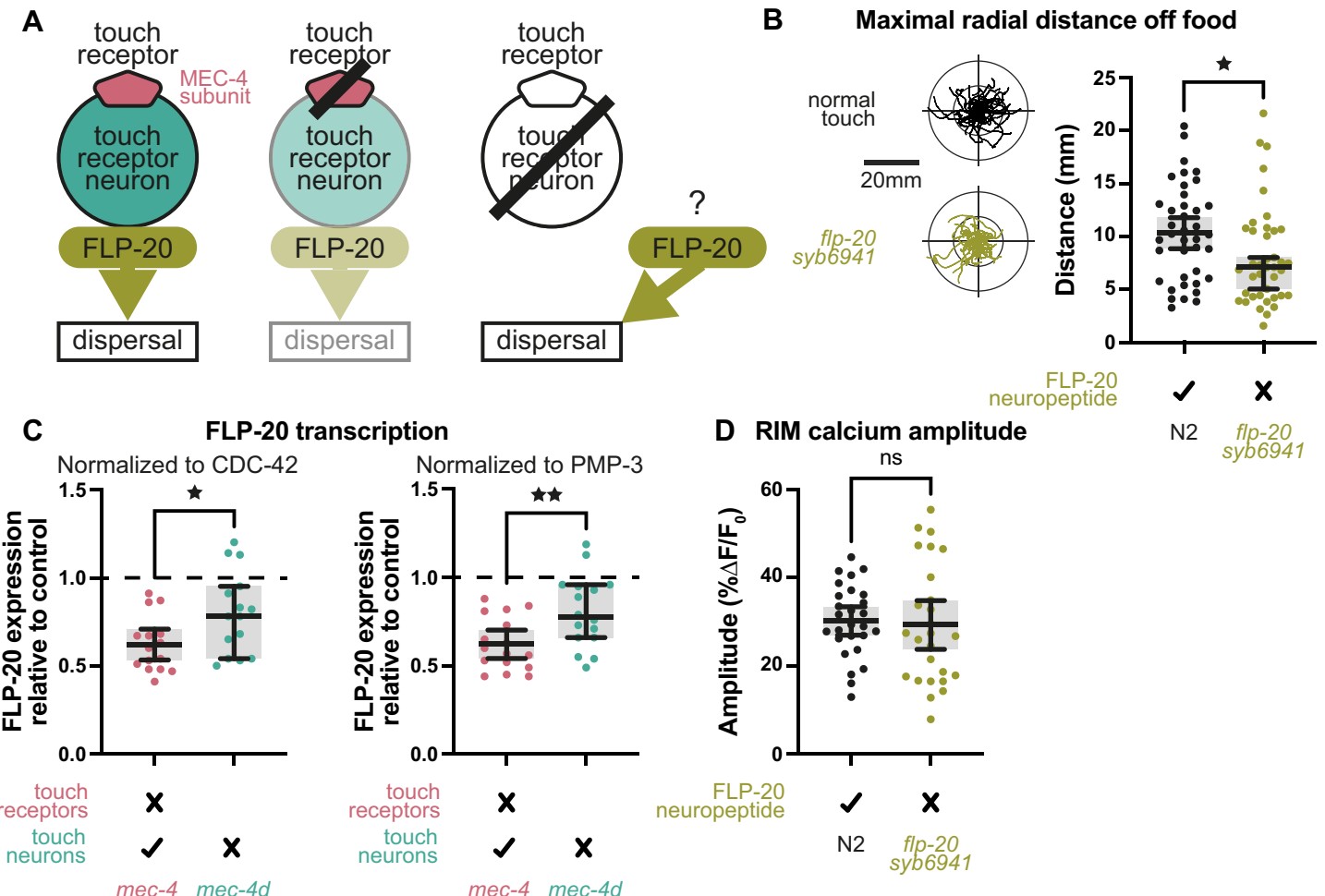

**Fig 4. FLP-20 transcription.** (A) Diagram comparing normal touch, loss of touch sensation, and loss of the TRNs, and the effects of FLP-20 neuropeptide signaling on dispersal. (B) Left, normal (N2) vs. *flp-20(syb6941)* mutant worm trajectories in the absence of food over a 2-min time window. Right, maximal radial distance from the start point of each trajectory. Mann–Whitney test. ★$p < 0.05$. $n = 39, 40$. (C) Quantitative reverse transcription PCR of FLP-20 transcripts normalized either to CDC-42 or PMP-3 housekeeping genes as expressed relative to normal touch-sensing worms (N2). *T* test ★$p < 0.05$, ★★$p < 0.01$. Left, $n = 15$. Right, $n = 16$. (D) RIM calcium amplitudes. *T* test. $N = 27, 26$. Bars indicate mean with 95% confidence intervals. The numerical data presented in this figure can be found in Supplementary file S1 Data.

[25]. To monitor FLP-20 expression in ASE, we used a transcriptional fusion of GFP to the FLP-20 promoter [25] and measured ASE fluorescence as an index of FLP-20 transcription in this neuron. Baseline FLP-20 expression was comparable in ASEL and ASER, the left and right neurons comprising the ASE neuron class, despite functional asymmetries that exist in this neuron pair [26] (S3C Fig). Previously, we found that FLP-20 expression in *mec-4* mutants is reduced both in the TRNs and in ASE [6]. We reproduced this result in a new experiment including also *mec-4d*, and observed that consistent with behavioral and RT-qPCR findings, *mec-4d* worms showed normal FLP-20 expression in ASE, unlike the decreased levels in *mec-4* (Fig 5A), suggesting that FLP-20 transcription in ASE is coupled to touch sensation in a TRN-dependent manner (Fig 4A). These results suggest that when touch sensation is compromised, some TRN signal may elicit a decrease in ASE FLP-20 expression, as a trigger for behavioral adjustment. However, if the TRNs are removed, this signal is eliminated and ASE FLP-20 levels remain normal (Fig 5C). To further establish this, we applied a site-directed recombination approach using Flp-FRT to eliminate FLP-20 expression exclusively in the ASE neurons [6] of *mec-4d* mutants (Fig 5B). This was sufficient to significantly decrease dispersal behavior even

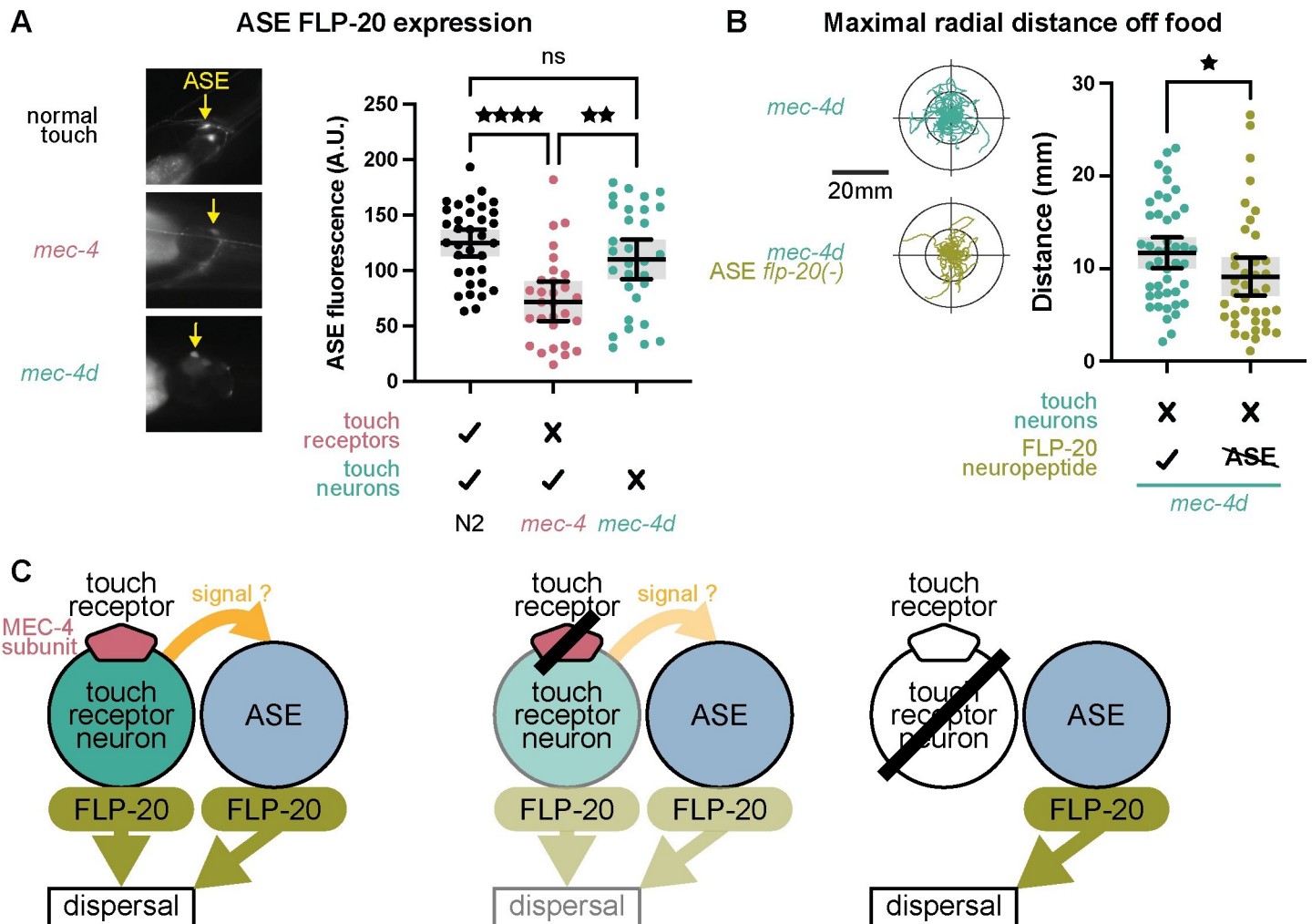

**Fig 5. FLP-20 expression in the ASE neurons is linked to touch sensitivity and touch neuron integrity.** (A) Left, example fluorescent images of the ASE neuron expressing the transcriptional fusion Pflp-20::GFP. ASE GFP fluorescence intensity indicates FLP-20 transcription levels. Right, mean ASE fluorescence within the ASE cell body after background subtraction. One-way ANOVA ($p < 0.0001$) followed by Tukey's multiple comparisons test. ★★$p < 0.01$, ★★★★$p < 0.0001$. $n = 34, 29, 30$. Bars indicate mean with 95% confidence intervals. (B) Left, *mec-4d* with normal vs. ASE-specific silenced FLP-20 worm trajectories in the absence of food over a 2-min time window. Right, maximal radial distance from the start point of each trajectory. Mann–Whitney test. ★$p < 0.05$. $n = 44, 39$. (C) Model illustrating the interplay between the TRNs and the ASE salt-sensing neurons in FLP-20 neuropeptide regulation of dispersal behavior. The numerical data presented in this figure can be found in Supplementary file S1 Data.

in worms lacking the TRNs. Eliminating FLP-20 from ASE in *mec-4d* mutants also resulted in increased reversing (S3D Fig). Together, these results suggest a model whereby normal touch sensation is accompanied by FLP-20 secretion from the TRNs and from the ASE neurons core-gulating dispersal behavior (Fig 5C). Loss of touch sensation reduces FLP-20 secretion from both the TRNs and ASE through some coordinating signal (Fig 5C), down-regulating dispersal behavior. Removal of the TRNs eliminates TRN FLP-20 secretion, but also the reduction in ASE FLP-20 secretion, thus maintaining dispersal at normal levels (Fig 5C).

## Additional TRN-expressed neuropeptides

Beyond FLP-20, additional neuropeptides are expressed in the TRNs (e.g., NLP-7 [27], FLP-4, and FLP-8 [25]) that might complement or, conversely, antagonize the effects of FLP-20 on

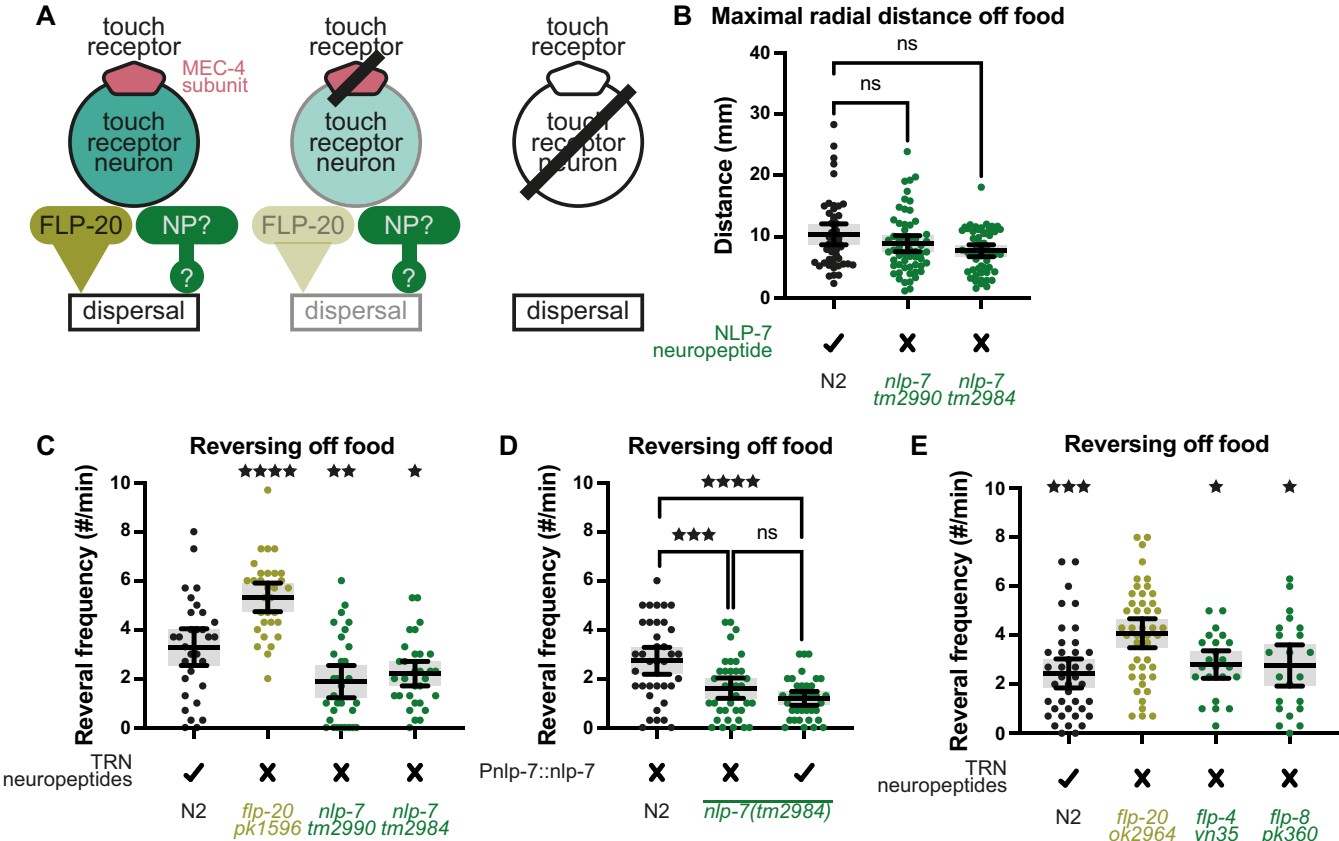

**Fig 6. Additional TRN-expressed neuropeptides.** (A) In parallel to FLP-20, other TRN-expressed neuropeptides (NP) might promote or suppress dispersal. (B) Maximal radial distance covered by each worm. Kruskal–Wallis test ($p = 0.101$), $n = 48, 49, 58, 54$. (C–E) Spontaneous reversing frequency in the absence of food. (C) One-way ANOVA ($p < 0.0001$) followed by Holm–Šídák's multiple comparisons test. ★★★★$p < 0.0001$, ★★$p < 0.01$, ★$p < 0.05$. $n = 31$ for each condition. (D) One-way ANOVA ($p < 0.0001$) followed by Holm–Šídák's multiple comparisons test. ★★★★$p < 0.0001$, ★★★$p < 0.001$. $n = 37$ for each condition. (E) One-way ANOVA ($p = 0.0004$) followed by Šídák's multiple comparisons test (no significant differences between N2 and *flp-4* or *flp-8*). ★★ ★$p < 0.001$, ★$p < 0.05$. $n = 40, 45, 23, 22$. Bars indicate mean with 95% confidence intervals. The numerical data presented in this figure can be found in Supplementary file S1 Data.

locomotion (Fig 6A). Interestingly, NLP-7 has previously been shown to be involved in suppressing ASH-mediated escape responses [28], reminiscent of the diminished response in *mec-4* mutants (Fig 2B). We thus examined dispersal behavior of *nlp-7* mutants. We expected that if NLP-7 suppresses dispersing, then these mutants should show elevated dispersal. Instead, we observed normal dispersal (Fig 6B). Surprisingly, at the same time, *nlp-7* reversal frequency was significantly decreased relative to normal worms (Fig 6C), exactly opposite to the increase in reversing of *flp-20* mutants (Fig 6C). This discrepancy could occur if the radial distance assay, while effective in detecting decreased dispersal (e.g., Fig 4B), might be insensitive to increased dispersing, due, for example, to a possible ceiling effect. Indeed, when we examined control N2 worms removed from their food for 1 min compared to 30 min, we found no difference in dispersal (S4A Fig), but a large and significant difference in reversing (S4B Fig). We therefore focused on reversal frequency analysis. We introduced into one of the *nlp-7* mutant strains a transgene containing functional *nlp-7* fused to its own promoter. This failed to rescue the decrease in reversal (Fig 6D), and thus we were not able to clearly establish a role for NLP-7 in behavioral adjustment. We therefore turned to test FLP-4 and FLP-8. The reversal frequency of these mutants was significantly smaller than that of *flp-20* mutants, and

similar to normal worms (Fig 6E), likely ruling out a role for these neuropeptides in TRN-related behavioral adjustment.

## Discussion

We have shown that loss of body touch sensation in *C. elegans* leads to a range of behavioral adjustments, which can be divided into 2 classes. First, cross-modal compensation, consisting of the increased performance of sensory modalities that are distinct from body touch, such as enhanced smell [6] or taste sensitivity, possibly providing an alternative channel for receiving information about the environment. Second, modality-specific behavioral restraint, exhibited by restrictive-like behaviors, such as confined dispersal, reduced escape following nose touch, and suppression of head oscillations during spontaneous reversing. These can be interpreted as preemptive measures for avoiding undetectable potential hazards that would normally be sensed by the touch system. Interestingly, and in line with this interpretation, similar cautiousness has been observed also in human gait, for example, following central visual field loss [29], or disruption of somatosensory input from the plantar afferents of the foot [30]. These findings highlight a role for sensory systems in permitting, under normal function, a degree of behavioral latitude, while reserving defensive behaviors as a response to specific sensory alerts. However, when such alerts become unavailable due to sensory loss, behavior switches to a default precautionary mode.

This putative principle of sensory operation is further emphasized by our finding that behavioral adjustment following loss of touch requires intact touch neurons. Although the removal of the touch neurons as in the *Pmec-17::ICE* strain (S1 Fig), or in the *mec-4(e1611)* mutant [13], eliminates touch sensation [31], we observed no behavioral adjustment in these strains, very different from *mec-4(u253)* touch receptor mutants. We found a corresponding pattern of selective modification in *mec-4*, but not *mec-4d* mutants in the activity of the RIM motor/interneuron, which plays an important and rather complex role in regulating locomotion [22,23].

Contrary to the lack of behavioral adjustment to the loss of touch, *mec-4(e1611)* mutants do exhibit altered locomotion [32–34] and mating behavior [35] when compared to normal worms. However, these behaviors are presumably directly touch dependent, as opposed to the behavioral modifications that we observe in *mec-4(u253)* worms, which appear to serve as adjustments to the indirect effects of loss of touch. Thus, for example, repetitive turning during mating of *mec-4(e1611)* males [35] is likely due to the lack of sensory input required for mating. In contrast, the escape response to nose touch (Fig 2B) does not require body touch input, but is repressed in *mec-4(u253)* mutants, perhaps because these worms are "blind" to potential obstacles that may be present in their backwards path, and so preemptively avoid them. This kind of adjustment occurs only if the worms are touch deficient and still possess intact touch neurons.

In what ways does the loss of touch neurons differ physiologically from the loss of touch receptor function alone? Previously, we have shown that the FLP-20 neuropeptide secreted by the touch neurons plays an important role in linking between touch input and the olfactory circuit [6], as well as the nose touch circuit [36]. We have demonstrated that loss of touch leads to reduced FLP-20 secretion, which in turn, enables stronger coupling between the AWC olfactory sensory neurons and the AIY interneurons [6]. We have now shown that eliminating FLP-20 leads also to restricted dispersal behavior as in touch-deficient worms, suggesting that FLP-20 signaling from the touch neurons is involved in regulating dispersal. We expected that the removal of the touch neurons would further decrease FLP-20 levels, leading to even more restricted dispersal. Instead, we found that the expression levels of FLP-20 are actually elevated

in *mec-4(e1611)* mutants compared to *mec-4(u253)*, with a specific increase in FLP-20 expression in the ASE gustatory neurons. It is unclear how the touch and ASE neurons coordinate their levels of FLP-20 expression. One possibility is that additional neuropeptides such as FLP-4, FLP-8, or NLP-7, while not necessarily directly involved in TRN-related behavioral modulation, could interact with the FLP-20 pathway, perhaps similarly to the interactions observed between other *C. elegans* neuropeptides [37,38]. Such additional signaling components could also underlie the changes in RIM neuronal activity that are not observed in the absence of FLP-20. Further investigation will be required to elucidate these points.

Another important difference between sensory neuron silencing, as in *mec-4(u253)*, compared to their removal, as in *mec-4(e1611)*, could be attributed to gap junction connectivity. As our previous analysis has shown [39], sensory neurons that are inactive may draw current away from electrically coupled partner neurons, decreasing their activation level, an effect that should not occur in the absence of the sensory neurons [39]. In *C. elegans*, the touch neurons are interconnected by gap junctions and form gap junction-based synaptic connections with premotor neurons, which drive the escape response to touch stimulation [9]. It is not unlikely that in *mec-4(u253)* mutants, the inactive touch neurons introduce a shunting effect that alters the activity of premotor neurons, impacting locomotion and other behaviors, whereas such shunting does not occur when the touch neurons are missing, as in *mec-4(e1611)* mutants. This possibility remains to be tested.

In conclusion, we have described cautious-like behaviors in touch-deprived *C. elegans* worms as a modality-specific form of behavioral adjustment to the loss of sensory input. We have shown that even in the absence of touch sensation, the touch neurons still play an important role in enabling neuronal and behavioral adjustment, and that the expression level of the neuropeptide FLP-20 in non-touch sensory neurons, ASE, correlates with the presence or lack of behavioral adjustment. These findings form the basis for the investigation of the molecular and cellular processes that underlie sensory neuron-dependent behavioral adjustment following sensory loss.

## Materials and methods

### *C. elegans* growth and maintenance

*C. elegans* strains were maintained under standard conditions at room temperature (22-23˚C) on nematode growth medium (NGM) 2% agar plates seeded with *Escherichia coli* strain, OP50. The N2 strain (Bristol, England) was used as the reference strain. All other strains are listed in Table 1.

### Behavioral analysis

For behavioral assays involving individual worms, young adults were picked manually. For population assays, worms were first age-synchronized by applying bleach solution (15% NaOCl and 10 M NaOH) to adult hermaphrodites, vortexing and washing the extracted eggs with M9 buffer, and then culturing the eggs in fresh NGM plates seeded with OP50 bacteria. Manual scoring of behavioral assays was performed such that the genotype of each worm was concealed from the experimenters during the experiment. Each experiment was repeated on at least 3 different days.

**Spontaneous reversing off food.**   A single worm was removed from a plate with food to an empty 6-cm NGM plate, allowed to crawl for a few seconds until no traces of food were visible in its track, and then transferred to another empty test plate. After 1 min, reversing events consisting of at least 1 body bend were counted over a 3-min period.

**Table 1.** *C. elegans* strains used in the study.

| Strain | Genotype | Description | Figures |
|---|---|---|---|
| TU253 | *mec-4(u253)* | Dysfunctional touch receptor | Figs 1, S1B, 2, S2A, 4C |
| CB1611 | *mec-4(e1611)* | Genetically ablated touch receptor neurons | Figs 1, S1A, S1C–S1E, 2, S2B, 4C |
| IRB14★ | *syIs595[Pmec-17::cGAL, Pofm-1::RFP]; sysIs414[15xUAS::let-858 3′ UTR, Punc-122::GFP]* | | S1A, S1C, S1D, S1E; S2B Figs |
| IRB32 | *mec-4(u253); syIs595; syIs414* | Touch receptor mutant with genetically ablated touch receptor neurons | S1B Fig |
| IRB33 | *mec-4(e1611); syIs595; syIs414* | Double genetically ablated touch receptor neurons | S1C Fig |
| CB1515 | *mec-10(e1515)* | Dysfunctional touch receptor | S2A Fig |
| IRB2041 | *yesEx42[Pcex-1::GCaMP6f::SL2::mCherry]* | RIM GCaMP6f calcium indicator | Fig 3 |
| IRB46 | *mec-4(u253); yesEx42* | | Fig 3 |
| IRB42 | *mec-4(e1611); yesEx42* | | Fig 3 |
| PHX6941 | *flp-20(syb6941)* | Lack of FLP-20 neuropeptide | Fig 4B |
| RB2188 | *flp-20(ok2964)* | Lack of FLP-20 neuropeptide | S3A Fig |
| IRB41 | *flp-20(syb6941); yesEx42* | RIM GCaMP6f calcium indicator | Fig 4D |
| NY2053 | *ynIs53[Pflp-20::GFP]* | ASE GFP transcriptional fusion | Figs S3C and 5A |
| IRB10 | *mec-4(u253); ynIs53* | | |
| IRB11 | *mec-4(e1611); ynIs53* | | |
| IRB43 | *mec-4(e1611); flp-20(syb6941); pekSi28[cb-unc-119(+)::Pflp-20::FRT::FLP-20::SL2::mCherry::terminator::FRT::GFP]II* | ASE-specific FLP-20 silencing | Figs 5B and S3D |
| IRB2043 | *mec-4(e1611); flp-20(syb6941); pekSi28; yesEx44[Pgcy-5::flpase, Pgcy-7::flpase, Punc54::mCherry]* | | |
| | *nlp-7(tm2990)* | Lack of NLP-7 neuropeptide | Figs 6A, 6B and S4 |
| | *nlp-7(tm2984)* | Lack of NLP-7 neuropeptide | Figs 6A–6C and S4 |
| PT505 | *flp-20(pk1596)* | Lack of FLP-20 neuropeptide | Fig 6B |
| IRB2048★★ | *nlp-7(tm2984); yesEx49 [Pnlp-7::nlp-7::nlp-7 3'UTR; Pmec-4::mCherry::unc-54 3'UTR]* | Attempted *nlp-7* rescue | Fig 6C |
| NY119 | *flp-4(yn35)* | Lack of FLP-4 neuropeptide | Fig 6D |
| PT501 | *flp-8(pk360)* | Lack of FLP-8 neuropeptide | Fig 6D |

★ Strain IRB14 was generated using the cGAL system [40], including a Pmec-17 driver strain (PS8373), a kind gift from Dr. Han Wang.

★★Strain IRB2048 was generated by injecting into *nlp-7(tm2984)* mutants a PCR-amplified segment from genomic DNA containing the *nlp-7* locus together with 2,557 bp upstream (promoter) and 1,431 bp downstream (3′ UTR) sequences as in [41]. The amplicon was injected at 50 ng/µl together with a Pmec-4::mCherry co-expression marker at 40 ng/µl.

**Odor chemotaxis.** Prior to the assay, we placed a 3 µl drop of 0.1% benzaldehyde (Bz) diluted in ethanol on one side of a 9-cm unseeded NGM test plate and a 3 µl drop of ethanol alone on the opposite end. To each drop, 1 µl drop of 1 M sodium azide (NaN$_3$) was added for trapping the worms. Approximately 200 synchronized young adult worms were washed with M9 buffer to remove bacteria and placed on the test plate. To avoid clumping of the worms (especially in the case of *mec-4(u253)* mutants, which tend to clump), worms were exposed to blue light for 1 min under a fluorescence microscope (Zeiss Stereo Discovery). Chemotaxis was scored 30 min later by counting the number of paralyzed worms within 3.5 cm from the edge of the plate to the center, on the side of the odor spot, N(odor) versus the side of the control spot, N(control), and calculating the chemotaxis index (CI) equal to: [N(odor)-N(control)]/[N(odor)+N(control)].

**Salt chemotaxis.** Salt chemotaxis was performed similarly to odor chemotaxis, except for stimulus application. Instead of a drop of odorant, we placed a salt-agarose plug (1 cm

diameter, 1 cm height, extracted from a plate containing 2 g agarose, 60 ml DDW, 40 ml NaCl 250 mM) on one side of the plate approximately16 h in advance and removed it just before the start of the experiment, applying 1 μl of 1 M sodium azide (NaN$_3$) on both ends of the plate. Chemotaxis was scored 2 h after placing washed worms on the plate and blue light exposure. The CI was calculated as in the odor chemotaxis assay.

**Dispersal off food.** Two young adult worms were removed from a plate with food to an empty 6-cm NGM plate, allowed to crawl for a few seconds until no traces of food were visible in their tracks, and then transferred to another empty plate. Test plates were placed under a camera (Pixelink), and 2-min long videos were captured. The videos were analyzed using TRex animal tracking system [42], and the maximal radial distance of each worm's path was calculated from the X-Y coordinates using a custom Matlab script (10.5281/zenodo.12193275) using a custom Matlab platform (10.5281/zenodo.12193290).

**Nose touch.** Nose touch assays were performed similarly to [43]. Test plates were prepared in advance by placing at the center of each plate a 20-μl drop of OP50, and leaving the plate to dry for 1 h with its lid open, creating in this way a thin bacterial lawn. Twelve worms were transferred to a single test plate and remained there for approximately 1 h before the start of the assay. An eyelash attached to a toothpick was placed just in front of each worm's trajectory, and the worm's response to its nose contacting the eyelash was scored as 0 –no response, 0.5 –just head withdrawal, or n–the number of body bends during reversal. This was repeated 5 times for each worm, rotating through all worms. The final score for each worm was the average over these 5 trials.

**Head oscillations.** High-resolution video excerpts of single worms obtained using Worm Tracker 2.0 [44] were examined by an experimenter. Head oscillations during several reversing events throughout each video were scored as either 0 (no oscillations), 1 (weak to moderate oscillations), and 2 (vigorous oscillations).

## Calcium imaging

Calcium imaging was performed using a microfluidic PDMS chip designed to trap *C. elegans* so that they can still move their body, but the head remains confined and in focus [24]. We placed the chip on top of a 40× air objective of a Zeiss AxioObserver 7 inverted fluorescence microscope connected to a Photometrics Prime 95B sCMOS camera. After 40 min of recording, each video was cropped to contain the imaged RIM neuron. The average raw fluorescence of the entire frame was derived using a custom written Matlab script [6]. Traces were then analyzed using an additional Matlab program [6], and the start and peak of calcium events throughout each trace were identified. The difference in fluorescence between them was used to calculate the amplitude of each event (in % values), and the amplitudes of the events along each trace were averaged. Thus, each data point represented the average RIM calcium event amplitude for a specific worm.

## Reverse transcription quantitative PCR

Synchronized eggs were placed on NGM plates seeded with OP50 *E. coli* bacteria. The worms were grown for 3 days. Young adult worms were then washed with M9 buffer to remove bacteria from the worms. Each worm pellet was resuspended in 1 M DTT and RA1 solution from the NucleoSpin1 RNA kit (Macherey-Nagel, Duren, Germany #740955.50), homogenized using a glass homogenizer (Dounce tissue grinder pestle, Merck P1110-1EA), and frozen at −80°C overnight. Samples were thawed in cycles of 20 s vortexing and a few seconds on ice. RNA extraction and purification were performed according to manufacturer instructions, and 1,000 ng of RNA was measured for cDNA production. cDNA was generated by reverse

transcription of the total RNA samples with a High-Capacity cDNA Reverse Transcription kit (Applied Biosystems, CA; #4368814) for RT-qPCR.

For *flp-20* qPCR, we used primers:
5′-TGGTTATCCTGGTCAAGAGC-3′
3′-TCATGTGGTTCATCTGTGCC-5′

qPCR was performed in triplicates using the Fast SYBR Green Master Mix (Applied Biosystems; #4385612) and quantified in a QS5 Real-Time PCR (96 wells) detection system (Applied Biosystems). The mRNA expression levels were separately normalized to the mRNA expression levels of 2 housekeeping genes, cdc-42, and pmp-3. Relative mRNA expression was calculated using the DDCT method.

## ASE-specific FLP-20 expression

To monitor ASE FLP-20 transcription, we recorded fluorescence intensity in ASE neurons of a *Pflp-20*::*GFP* integrated transgene. Transgenic animals were picked at the young adult stage to 5% agarose slides, treated with 1 M sodium azide (NaN$_3$) for immobilization and then imaged in an inverted fluorescence microscope (Zeiss AxioObserver 7) using 32× (20× objective and 1.6× digital zoom) magnification with fixed exposure time and LED intensity. Raw grayscale JPG images were analyzed using a custom Matlab script (10.5281/zenodo.12193342) on a custom Matlab platform (10.5281/zenodo.12193290). For each image, a rectangular region of interest (ROI) was manually adjusted around the neuron cell body. If 2 neurons were visible, then the clearer/brighter of the 2 was chosen. The image within the ROI was cropped and binarized according to the Otsu method (Matlab). The number of on-pixels was considered as the area of the cell body. The sum of the on-pixel intensities was divided by the area to derive the average fluorescence intensity. The same ROI was positioned over a non-fluorescent region within the same worm, and an equivalent area was averaged for pixel intensity and used as the background value, which was subtracted from the main value. One-way ANOVA showed no significant differences between the areas or background values of the different strains.

## Statistical analysis

Statistical analysis was performed using Prism GraphPad. For each dataset, initial normality tests were carried out in order to determine whether to perform a parametric or nonparametric test. Details about sample sizes and the statistical tests applied are provided in the figure legends. Raw data are provided in the Supporting Data File.

## Supporting information

**S1 Data. Numerical values of all data presented in all figure.**
(XLSX)

**S1 Fig. Reversing and chemotaxis following TRN genetic ablation.** (A–C) Spontaneous reversing frequency in the absence of food. (A) Šídák's test, $n = 15$ for each condition. (B) Kruskal–Wallis test ($p < 0.0001$) followed by Dunn's multiple comparisons test. ★★$p < 0.01$. $n = 19$ for each condition. (C) One-way ANOVA ($p = 0.2$). $n = 18$ for each condition. (D) Chemotaxis to 1:1,000 Benzaldehyde (Bz). Dunn's test, $n = 27$ for each condition. (E) Chemotaxis to 100 mM NaCl. Dunn's test, $n = 28, 35$. The data presented in panels A, D, and E was obtained in the same experiments as shown in Fig 1B–1D, respectively. The numerical data presented in this figure can be found in Supplementary file S1 Data.
(TIF)

**S2 Fig. Dispersal following TRN receptor deficiency and genetic ablation.** (A, B) Left, aligned worm trajectories in the absence of food over a 2-min time window. Right, maximal radial distance of worm trajectories. (A) One-way ANOVA ($p < 0.0001$) followed by Tukey's multiple comparison test. ★$p < 0.05$, ★★$p < 0.01$, ★★★★$p < 0.0001$. $n = 32, 30, 25$. (B) Dunn's test, $n = 76, 77$. The data presented in this plot was obtained in the same experiment as shown in Fig 2A. Bars indicate mean with 95% confidence intervals. The numerical data presented in this figure can be found in Supplementary file S1 Data.
(TIF)

**S3 Fig. flp-20 and ASE.** (A) Left, normal vs. *flp-20(ok2964)* mutant trajectories in the absence of food over a 2-min time window. Right, maximal radial distance from the start point of each trajectory. Mann–Whitney test. ★★★$p < 0.001$. $n = 46,62$. (B) *flp-20* RT-qPCR primers (5′-TGGTTATCCTGGTCAAGAGC-3′; 3′-TCATGTGGTTCATCTGTGCC-5′) produce an expected specific 152-bp long genomic amplicon. (C) Mean ASEL vs. ASER fluorescence within the ASE cell body after background subtraction. Mann–Whitney test. $n = 27, 14$. (D) Reversing frequency off food of *mec-4d* with normal vs. ASE-specific silenced FLP-20. Mann–Whitney test. ★$p < 0.05$. $n = 40$. Bars indicate mean with 95% confidence intervals. The numerical data presented in this figure can be found in Supplementary file S1 Data.
(TIF)

**S4 Fig. Ceiling effect of dispersal assay.** (A, B) Comparison between worms that spent 1 min vs. 30 min off food before the start of the assay. (A) Maximal radial distance from the start point of each trajectory, Mann–Whitney test ($p = 0.503$). $n = 17, 18$. (B) Reversing frequency off food, Mann–Whitney test. ★★★★$p < 0.0001$. $n = 15$ for each condition. Bars indicate mean with 95% confidence intervals. The numerical data presented in this figure can be found in Supplementary file S1 Data.
(TIF)

## Acknowledgments

We are grateful to the *Caenorhabditis elegans* Genetic Consortium (funded by NIH Office of Research Infrastructure Programs P40 OD010440), The National BioResource Project, Tokyo Women's Medical University, Tokyo, Japan, and Dr. Han Wang for *C. elegans* strains. We thank Dr. Catharine Rankin and Alex Yu for valuable feedback on the study, and Hana Boocholez-Vardi for technical assistance.

Views and opinions expressed in this article are those of the authors only and do not necessarily reflect those of the European Union or European Innovation Council and SMEs Executive Agency (EISMEA). Neither the European Union nor the granting authority can be held responsible for them.

## Author Contributions

**Conceptualization:** Michal Staum, Ayelet-Chen Abraham, Ithai Rabinowitch.

**Data curation:** Michal Staum, Reema Arbid, Varun Sanjay Birari, Matanel Dominitz.

**Formal analysis:** Michal Staum, Ayelet-Chen Abraham, Reema Arbid, Varun Sanjay Birari, Ithai Rabinowitch.

**Funding acquisition:** Ithai Rabinowitch.

**Investigation:** Michal Staum, Ayelet-Chen Abraham, Reema Arbid, Varun Sanjay Birari, Matanel Dominitz, Ithai Rabinowitch.

**Methodology:** Michal Staum, Ayelet-Chen Abraham, Reema Arbid, Varun Sanjay Birari, Matanel Dominitz, Ithai Rabinowitch.

**Project administration:** Ithai Rabinowitch.

**Software:** Ithai Rabinowitch.

**Supervision:** Ithai Rabinowitch.

**Writing – original draft:** Ithai Rabinowitch.

**Writing – review & editing:** Ithai Rabinowitch.

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
