## [Decision Letter · Decision Letter 0]

21 Dec 2023

Dear Dr Rabinowitch, 

Thank you for submitting your manuscript entitled "Behavioral adjustment to mechanosensory loss requires intact mechanosensory neurons" for consideration as a Update Article by PLOS Biology. I am contacting you in the absence of my colleague Christian Schnell from the office. 

Your manuscript has now been evaluated by the PLOS Biology editorial staff, an academic editor with relevant expertise and is under consideration by one reviewer. However, before we can make a decision on your work, we need you to complete your submission by providing the metadata associated with it. 

To this end, please login to Editorial Manager where you will find the paper in the 'Submissions Needing Revisions' folder on your homepage. Please click 'Revise Submission' from the Action Links and complete all additional questions in the submission questionnaire.

Once your full submission is complete, your paper will undergo a series of checks. After your manuscript has passed the checks it will be back with Christian Schnell, who will be in touch as soon as we are in a position to make a decision, likely early in January. 

To provide the metadata for your submission, please Login to Editorial Manager (https://www.editorialmanager.com/pbiology) within two working days, i.e. by Dec 23 2023 11:59PM.

Kind regards,

Nonia

Nonia Pariente, PhD

Editor in Chief

PLOS Biology

on behalf of

Christian Schnell, PhD

Senior Editor

PLOS Biology

cschnell@plos.org

---

## [Editor Report · Decision Letter 1]

11 Jan 2024

Dear Dr Rabinowitch,

Thank you for your patience while we considered your revised manuscript "Behavioral adjustment to mechanosensory loss requires intact mechanosensory neurons" for publication as a Update Article at PLOS Biology. Your revised study has been evaluated by the PLOS Biology editors, the Academic Editor and one of the original reviewers.

In light of the reviews, which you will find at the end of this email, we would like to invite you to revise the work to thoroughly address the reviewers' reports one final time. While we think the paper ahs been strengthened, Reviewer 1 has a number of lingering concerns that we think would need to be thoroughly addressed for the paper to meet our bar. We realize that this would likely require a substantial amount of additional work and we would understand if you wish to pursue faster publication elsewhere. We are more than happy to facilitate portable peer review if you think that would be helpful. 

If you wish to submit a revised manuscript, please note that we will aim to make a final decision after the next round of review. We strongly encourage you to send us a revision plan (leaving aside the comment about this being mechanistically as detailed as your previous paper) which we can discuss with the Academic Editor, to hopefully help to avoid unnecessary work and expenses. 

Given the extent of revision needed, we cannot make a decision about publication until we have seen the revised manuscript and your response to the reviewers' comments. Your revised manuscript is likely to be sent for further evaluation by all or a subset of the reviewers.

**IMPORTANT - SUBMITTING YOUR REVISION**

*Re-submission Checklist*

*Published Peer Review*

*PLOS Data Policy*

*Blot and Gel Data Policy*

Sincerely,

Christian

Christian Schnell, PhD

Senior Editor

PLOS Biology

cschnell@plos.org

REVIEWS:

The authors have included some new data, but they fall short of addressing my major concerns. For example, they imaged RIM neuron activity, but this is insufficient to explain the observed differences in mec-4 and mec-4d worms. They should image sensory neurons (ASE and AWC) and additional interneurons (AIB, AIY and AVA) in wt, mec-4 and mec-4d worms as they did in their previous paper. For the newly reported "restrictive-like behaviors", they should at least image ASH neurons that sense nose touch stimuli. In their previous paper, they provided a nice explanation to how FLP-20 released by TRNs may inhibit AWC-AIY information flow, thereby inhibiting chemotaxis behavior. But this mechanism cannot explain their current observations. So what is the mechanism? Overall, the work is interesting but rather descriptive and superficial and lacks depth. I would like to see the kind of mechanistic study the authors accomplished in their previous work published in PLoS Biology in 2016.

---

## [Editor Report · Decision Letter 2]

20 Jun 2024

Dear Ithai,

Thank you for your patience while we considered your revised manuscript "Behavioral adjustment to mechanosensory loss requires intact mechanosensory neurons" for publication as a Update Article at PLOS Biology. This revised version of your manuscript has been evaluated by the PLOS Biology editors and the Academic Editor.

Based on our Academic Editor's assessment of your revision, we are likely to accept this manuscript for publication, provided you satisfactorily address the following data and other policy-related requests:

* We would like to suggest a different title to improve readability: "Behavioral adjustment of C. elegans to mechanosensory loss requires intact mechanosensory neurons"

* DATA POLICY:

Regardless of the method selected, please ensure that you provide the individual numerical values that underlie the summary data displayed in the following figure panels as they are essential for readers to assess your analysis and to reproduce it: 1BCD, 2ABC, 3B, 4BCD, 5AB, 6BCDE, S1, S2, S3 and S4.

* CODE POLICY

We expect to receive your revised manuscript within two weeks. 

*Published Peer Review History*

*Press*

Sincerely,

Christian

Christian Schnell, PhD

Senior Editor

cschnell@plos.org

PLOS Biology

---

## [Editor Report · Decision Letter 3]

2 Jul 2024

Dear Ithai,

Thank you for the submission of your revised Update Article "Behavioral adjustment of C. elegans to mechanosensory loss requires intact mechanosensory neurons" for publication in PLOS Biology. On behalf of my colleagues and the Academic Editor, Piali Sengupta, I am pleased to say that we can in principle accept your manuscript for publication, provided you address any remaining formatting and reporting issues. These will be detailed in an email you should receive within 2-3 business days from our colleagues in the journal operations team; no action is required from you until then. Please note that we will not be able to formally accept your manuscript and schedule it for publication until you have completed any requested changes.

PRESS

Sincerely, 

Christian

Christian Schnell, PhD

Senior Editor

PLOS Biology

cschnell@plos.org